## [Editor Report · Decision Letter 0]

19 Sep 2025

Dear Dr Liang,

Thank you for submitting your manuscript entitled "Fine-tuning ERK activity enables proliferation-differentiation balance during lineage specification" for consideration as a Research Article by PLOS Biology.

Your manuscript has now been evaluated by the PLOS Biology editorial staff as well as by an academic editor with relevant expertise and I am writing to let you know that we would like to send your submission out for external peer review.

Once your full submission is complete, your paper will undergo a series of checks in preparation for peer review. After your manuscript has passed the checks it will be sent out for review. To provide the metadata for your submission, please Login to Editorial Manager (https://www.editorialmanager.com/pbiology) within two working days, i.e. by Sep 23 2025 11:59PM.

Kind regards,

Ines

--

Ines Alvarez-Garcia, PhD

Senior Editor

PLOS Biology

---

## [Decision Letter · Decision Letter 1]

21 Nov 2025

Dear Dr Liang,

Thank you for your patience while your manuscript entitled "Fine-tuning ERK activity enables proliferation-differentiation balance during lineage specification" was peer-reviewed at PLOS Biology. It has now been evaluated by the PLOS Biology editors, an Academic Editor with relevant expertise, and by two independent reviewers.

The reviews are attached below. As you will see, both reviewers find the conclusions interesting, but they also raise several points that would need to be addressed before we can consider the manuscript for publication. Reviewer 1 asks for several clarifications and the addition of a figure showing the workflow for the single-cell tracking. Reviewer 2 thinks you should perform ERK/ELK ChIPseq in ME, instead of ESCs, given that chromatin landscapes and ERK activity are different and that this could lead to overinterpretation of the data. In addition, the reviewer asks for several clarifications and further discussion on the potential mechanisms by which ERK specifically promotes the translational rate of ME-associated proteins.

In light of the reviews, we would like to invite you to revise the work to thoroughly address the reviewers' reports. Please note that we cannot make a decision about publication until we have seen the revised manuscript and your response to the reviewers' comments. Your revised manuscript is likely to be sent for further evaluation by all or a subset of the reviewers.

**IMPORTANT - SUBMITTING YOUR REVISION**

3. Resubmission Checklist

a) *PLOS Data Policy*

b) *Published Peer Review*

Sincerely,

Ines

--

Ines Alvarez-Garcia, PhD

Senior Editor

PLOS Biology

Reviewers' comments

Rev. 1: Marta Markiewicz-Potoczny - please note that this reviewer has signed the review

In the manuscript "Fine-tuning ERK activity enables proliferation-differentiation balance during lineage specification" Ma and colleagues investigate the quantitative regulation of proliferation and differentiation during lineage development by ERK. The authors focus on understanding the mechanisms that fine-tune decision-making during development, and how and when a cell decides whether to commit to proliferation or differentiation. It is known that growth factors and morphogens manage both proliferation and differentiation toward development, but the exact mechanisms underlying this control are not well understood. ERK (FGF/MEK/ERK) signaling pathway is a key regulator of proliferation and cell fate specification, yet, as the authors note, the dynamic signaling patterns of ERK during lineage specification are not well defined, as is it clear how ERK simultaneously controls both processes.

In this manuscript the authors address an important question: how does ERK control proliferation and differentiation in lineages characterized by elevated or diminished ERK activity? They specifically examine the mesendoderm (ME) specification, which is associated with high ERK activity, using multiplex quantitative live-cell imaging and ERK titration with two ERK inhibitors: Ulixertinib and PD0325901. Strikingly, the authors observed that differentiation and proliferation occur at distinct ranges of ERK activity levels. Mechanistically they show that transcriptional and translational responses exhibit gene-specific sensitivity to ERK.

Collectively, this manuscript provides important insights into how proliferation and differentiation are controlled to enable developmental diversity. The authors combine live-cell imaging with quantitative modulation of ERK activity, revealing changes in ERK dynamics and their downstream targets. The manuscript is well written, and a subset of data presented is of high quality. However, this reviewer has several minor points that need further clarification.

Line 83: Please change "a simultaneously control" to "a simultaneous control".

Lines 187-190: "Higher ERK activity tended to associate with a higher TBXT expression (…)". This result is expected, and the authors' observations confirm previously reported phenomena that higher ERK activity spatially and temporally overlaps with TBXT expression. This sentence should be revised accordingly, and the work of others should be referenced, examples may include: PMID: 39026750, PMID: 39415005, PMID: 24445144, PMID: 37590131.

Lines 273-292: Can the authors speculate, perhaps based on Figure 5E and GO, how ERK might enhance translation efficiency of TBXT and other ME proteins?

Line 477: "Clones bearing the correct reporter expression, karyotype, differentiation potential were used for subsequent experiments" - How did the authors validate those characteristics? What does the "correct reporter expression" mean in this context?

Lines 492-509: A figure showing the workflow for the single-cell tracking, including example images illustrating each step, would greatly improve clarity for readers.

Lines 510-528: The raw data set should be provided as supplementary material. How many raw images were collected per genotype per condition per experiment? How does the exclusion of four frames during nuclear membrane breakdown affect quantification results?

Line 117: A typo "More 410 single cells were tracked (…)".

Figures:

Figure 1:

D-C - It will be clearer for the reader if panel D was placed directly below panel C.

G - How was the cell cycle measured and plotted? This question applies to all the other graphs presented in the manuscript (e.g. Figure S2H).

Figure 3:

B - The color-code seems inconsistent, the black control is missing from the legend. Also, it is difficult to distinguish TBXT intensity across Ulixertinib concentrations.

E-F - Please, re-order the graphs for logical flow.

H - it is unclear how the graph was generated and what it shows. Including example images would help readers interpret the data.

Figure 5:

C - Figure legend refers to "The dotted blue line is the trend line (…)" - but no such line can be seen in the graph. Please, revise or clarify.

Figure S1:

A - Nuclear morphology differs between images (e.g. 0H vs 6h, 6h vs 10h) yet the scale bar remains constant. Please, verify consistency.

B - TBXT expression after 12 h of differentiation (the first time point) is already high and comparable to later time points, which mildly contrasts with Figure 1B where early expression is low. Authors should present data from the pre-differentiation stage (t = 0 h).

H - The GAPDH blot appears unusual compared to others in the manuscript; please verify image integrity.

Rev. 2:

In this study, Ma and colleagues aim to dissect ERK's functional mechanisms in regulating cell proliferation and lineage specification in human mesendoderm (ME), a context in which elevated ERK activity is essential for differentiation. The authors generated a spectrum of ERK activity using ERK and MEK inhibitors and monitored the dynamics of TBXT (a ME marker) expression, ERK activity, and cell cycle progression in live cells. Notably, they demonstrate that ME-related and cell-cycle-related genes exhibit distinct transcriptional and translational sensitivities to ERK activity. Overall, this study provides rigorous side-by-side comparisons of ERK-dependent differentiation and proliferation responses within the same cellular contexts and offers a quantitative assessment of ERK's contributions to these key processes. I offer the following comments to improve the clarity and strengthen the mechanistic insights of the study.

Comments:

* More detailed and explicit explanations are needed for the analyses of translational efficiency. For example, what does each "ribosome fraction" represent in terms of translational activities? What exactly is measured by the HGP incorporation assay?

* The use of ERK/ELK ChIP-seq data generated in ESCs, rather than in ME cells, is not appropriate for drawing mechanistic conclusions. Because chromatin landscapes and ERK activities differ substantially between ESCs and ME, these ESC-derived binding profiles may not reflect regulatory interactions in ME. The authors should avoid over-interpreting these data, and performing ERK/ELK ChIP-seq in ME is recommended.

* The manuscript would benefit from discussion of potential mechanisms by which ERK specifically promotes the translational rate of ME-associated proteins.

* In Figure 5G, the minimum value of the Y-axis in the TPM plots should be set to 0 to allow accurate interpretation of expression changes.

---

## [Editor Report · Decision Letter 2]

18 Feb 2026

Dear Dr Liang,

Thank you for your patience while we considered your revised manuscript entitled "Fine-tuning ERK activity enables proliferation-differentiation balance during lineage specification" for publication as a Research Article at PLOS Biology. This revised version of your manuscript has been evaluated by the PLOS Biology editors and the Academic Editor.

Based on our Academic Editor's assessment of your revision, we are likely to accept this manuscript for publication, provided you satisfactorily address the data and other policy-related requests stated below my signature.

In addition, we would like you to consider a suggestion to improve the title:

"Fine-tuning ERK activity enables proliferation-differentiation balance during lineage specification of human pluripotent stem cells"

We expect to receive your revised manuscript within two weeks.

*Published Peer Review History*

*Press*

Sincerely,

Ines

--

Ines Alvarez-Garcia, PhD

Senior Editor

PLOS Biology

DATA POLICY:

Many thanks for providing the data underling the graphs shown in the figure. I have checked all the data and I would like you to clarify two points:

- Fig. 3F data seems to be missing. Please provide it or indicate where can be found.

- Please check if the data in for Fig. S3Q and S3R are correct or they are swapped.

In addition, please make the data you have deposited in the GEO database (GSE273286) publicly available at this stage.

---

## [Editor Report · Decision Letter 3]

2 Mar 2026

Dear Dr Liang,

Thank you for the submission of your revised Research Article entitled "Fine-tuning ERK activity enables proliferation-differentiation balance during lineage specification of human embryonic stem cells" for publication in PLOS Biology. On behalf of my colleagues and the Academic Editor, Christa Buecker, I am delighted to let you know that we can in principle accept your manuscript for publication, provided you address any remaining formatting and reporting issues. These will be detailed in an email you should receive within 2-3 business days from our colleagues in the journal operations team; no action is required from you until then. Please note that we will not be able to formally accept your manuscript and schedule it for publication until you have completed any requested changes.

PRESS

Sincerely,

Ines

--

Ines Alvarez-Garcia, PhD

Senior Editor

PLOS Biology
